# Probiotics, prebiotics and synbiotics for chronic kidney disease: protocol for a systematic review and meta-analysis

Ming Pei,[1] Lijuan Wei,[2] Shouci Hu,[2] Bo Yang,[1] Jinhua Si,[3] Hongtao Yang,[1] Jingbo Zhai[4]

MP and LW contributed equally.

[1]Department of Nephrology, First Teaching Hospital of Tianjin University of Traditional Chinese Medicine, Tianjin, China
[2]Graduate College, Tianjin University of Traditional Chinese Medicine, Tianjin, China
[3]Library, Tianjin University of Traditional Chinese Medicine, Tianjin, China
[4]Institute of Traditional Chinese Medicine, Tianjin University of Traditional Chinese Medicine, Tianjin, China

**Correspondence to**
Dr Hongtao Yang;
tjtcmht@126.com and Dr Jingbo Zhai;
zhaijingbo@foxmail.com

## ABSTRACT

**Introduction**  There is a growing interest in probiotic, prebiotic and synbiotic supplements for patients with chronic kidney disease (CKD). However, a systematic review and evaluation is lacking. The purpose of the present study is to assess the efficacy and safety of probiotics, prebiotics and synbiotics for non-dialysis or non-renal transplant patients with CKD.

**Methods and analysis**  An extensive literature search will be undertaken to identify potentially eligible studies from electronic databases including PubMed (1946 to present), EMBASE (1974 to present), Web of Science (1900 to present) and the Cochrane Central Register of Controlled Trials (CENTRAL, all years). No language restriction will be applied to the search. Both parallel and crossover randomised controlled trials will be included. The risk of bias of each included study will be assessed using the Cochrane Risk of Bias Tool. The primary outcome measures are uraemic toxins. Secondary outcomes include kidney function, adverse cardiovascular events, all-cause mortality, cause-specific death, progression to end-stage kidney disease, quality of life, gastrointestinal function and adverse events. Data will be synthesised using appropriate statistical methods. The quality of evidence for each outcome will be assessed using the Grading of Recommendations Assessment, Development and Evaluation approach.

**Ethics and dissemination**  No ethical approval is required as no primary data will be collected. We will publish findings from this systematic review in a peer-reviewed scientific journal, and the data set will be made freely available.

**PROSPERO registration number**  CRD42017079177.

## BACKGROUND

Chronic kidney disease (CKD) is one of the most common chronic diseases worldwide.[1] According to the Kidney Disease: Improving Global Outcomes (KDIGO) Guideline 2012,[2] CKD is defined as abnormalities of kidney structure or function, present for more than 3 months, with implications for health and is categorised into five stages according to severity using estimated glomerular filtration rate (GFR) or into three stages based on albuminuria.[3]

### Strengths and limitations of this study

► This is the first systematic review to investigate the efficacy and safety of probiotics, prebiotics or synbiotics for non-dialysis or non-renal transplant patients with chronic kidney disease (CKD).
► The search strategy was developed by a medical librarian with 10 years of experience as an information specialist.
► This protocol was developed following the Preferred Reporting Items for Systematic Review and Meta-Analysis Protocols 2015 statement.
► Study heterogeneity is likely to pose challenges for this meta-analysis.

The incidence of CKD is subject to variation worldwide.[4] The global prevalence of CKD is estimated to vary between 8% and 16%, and the number is 10.8% in China and 13.1% in the USA.[5 6] Diabetes and hypertension are the main risk factors for CKD.[4] The global prevalence of CKD is likely to increase in view of the growing number of patients with diabetes and/or hypertension.

Under normal conditions, homeostasis is maintained owing to interactions between the host and intestinal microflora.[7] However, quantitative and qualitative alterations in the intestinal microflora are present in patients with CKD.[7] This dysbiotic intestinal microflora is characterised by an increase in the pathogenic flora relative to the symbiotic flora.[7] The pathogenic gut microbiota produce uraemic toxins, in particular indoxyl sulfate (IS) and p-cresyl sulfate (PCS) among others, which have been associated with increased inflammation, greater oxidative stress[7 8] and higher risk for cardiovascular disease (CVD), progression of CKD and death due to CKD.[9 10]

Probiotics refer to the living microorganisms which colonise or implant in the host's gastrointestinal (GI) environment and exert beneficial health effects.[11 12] Prebiotics are

defined as non-digestible food ingredients that induce the growth and/or activity of beneficial microorganisms in the host.[11 12] Synbiotics are a mixture of probiotics and prebiotics.

Recently, probiotics, prebiotics or synbiotics have been reported to reduce inflammation, improve kidney function and retard progression of CKD by restoring the symbiosis of gut microflora in patients with CKD. A randomised trial found synbiotics decreased serum PCS without reducing serum IS in non-dialysis CKD.[13] A pilot study suggested probiotic dietary supplements are more effective than placebo in reducing blood urea nitrogen (BUN) and improving the quality of life of patients with stage 3 or 4 CKD.[14] Another study found that synbiotics delayed CKD progression.[9]

Guldris et al performed a narrative review of gut microbiota in relation to chronic kidney disease in 2017.[7] Their findings are open to dispute due to evident methodological flaws. First, literature retrieval was insufficient, largely because a comprehensive search strategy was not established. Second, the quality of the included studies was not assessed against a validated tool. Third, no statistical analysis was done and the evidence was left ungraded.

Another systematic review found prebiotic and probiotic therapies reduced IS and PCS in patients with end stage kidney disease (ESKD) on haemodialysis.[15] However, it is unclear whether the results hold true for other patients with CKD. To our knowledge, previous systematic reviews have not fully addressed the research topic in question.

The objective of this study is to provide a comprehensive systematic review of the available evidence on the efficacy and safety of probiotics, prebiotics and synbiotics for the management of non-dialysis or non-renal transplant patients with CKD.

## METHODS
This protocol was developed following the Preferred Reporting Items for Systematic reviews and Meta-Analyses guidelines for protocols (PRISMA-P).[16]

## Types of studies
We will include parallel and crossover randomised controlled trials (RCTs). Quasi-RCTs, controlled clinical trials (CCTs), controlled before and after trials and cluster-RCTs will be excluded to minimise biased estimates of treatment effects.[12]

## Types of participants
Non-dialysis or non-renal transplant patients at any stage of CKD (as defined by KDIGO) will be included.[17] We will consider patients of any age, both sexes, any ethnicity and studies in any clinical setting. Trials including healthy people or patients without CKD in the control group will be excluded.

## Types of interventions
### Experimental interventions
Probiotics, prebiotics and synbiotics regardless of dose, frequency, duration or route of delivery and in combination or as the only preparation will be included.[18]

### Comparator interventions
Placebo, no treatment or active pharmacological or non-pharmacological treatment (regardless of dose, frequency, duration, route of delivery or setting, and in combination or as the only preparation) will be included. We will exclude RCTs with probiotics, prebiotics and synbiotics in both arms.

## Types of outcome measures
### Primary outcome
The primary outcome measures are uraemic toxins (including but not limited to phenols and indoles). Phenols include p-cresol, PCS and p-cresyl glucuronide.[7] Indoles include IS and indoleacetic acid.[7]

### Secondary outcomes
The secondary outcome measures are listed below:
► Kidney function measures (including but not limited to BUN, GFR, creatinine clearance and serum creatinine).
► Major adverse cardiovascular events as defined by the investigator (including but not limited to myocardial infarction, coronary artery disease, heart failure, cerebrovascular disease and peripheral vascular disease).
► All-cause mortality and cause-specific death, such as cardiovascular mortality, sudden death and infection-related mortality.
► Progression to ESKD requiring renal replacement therapy (RRT: haemodialysis, peritoneal dialysis or kidney transplantation).
► Quality of life measured by a validated scale, such as the Kidney Disease Quality of Life.
► GI function (including but not limited to improvement in GI symptoms, transit time and tolerance).
► Adverse events.

## Search methods for identification of studies
### Electronic searches
An extensive literature search will be undertaken to identify potentially eligible studies in electronic databases including PubMed (1946 to present), EMBASE (1974 to present), Web of Science (1900 to present) and the Cochrane Central Register of Controlled Trials (CENTRAL, all years).

The search strategy was developed by a medical librarian (JS) with 10 years of experience as an information specialist, taking into consideration of the search terms used in previous reviews.[17 18] The search strategy for the PubMed database is provided in online Supplementary Appendix 1. The search terms are adapted properly to cater to each database. No language restriction will be applied to the search.

## Searching other resources

We will search ClinicalTrials.gov and the WHO International Clinical Trials Registry Platform for relevant unpublished or ongoing studies. Google Scholar will be searched using key words to identify grey literature.[19] The reference lists of all retrieved studies and previous systematic reviews will be checked.

## Data collection and analysis

### Selection of studies

Two review authors (LW and BY) will independently run all the literature searches. The identified references from electronic database searching and other resources will be imported into the EndNote software.[20] Duplicate records will be removed. The title and abstract of each citation will be screened to identify potentially eligible studies, followed by full-text review to confirm inclusion. Any disagreement will be resolved by consensus or consultation with a third reviewer (JZ).

Including duplicated data in a meta-analysis may lead to overestimation of the intervention effects.[21] If uncertainties remain in justifying multiple publications from a single data set, the author of the original reports will be contacted for clarification.[21]

A PRISMA flow diagram will be provided to state the process of study selection.

### Data extraction and management

Two review authors (SH and HY) will independently extract data from included studies using a standardised data extraction form. The form will be specially designed and piloted by the responsible reviewer. Any disagreement will be resolved through discussion. A third reviewer (JZ) will be consulted for a final decision if consensus cannot be reached.
► Study characteristics: title, author, publication year, design, sample size, funding source and use of randomisation, allocation concealment, blinding and control.
► Participant characteristics: age, sex, number in each group and CKD diagnostic criteria.
► Intervention details: intervention treatment, comparator treatment, dose, route of administration and number of cases included in statistical analysis.
► Outcome measures reported regarding efficacy and safety.

### Assessment of risk of bias for included studies

Two review authors (MP and JS) will independently assess the risk of bias for each included study using the Cochrane Risk of Bias Tool in RevMan V.5.3.[21]

The methodological quality of included studies will be assessed from seven aspects, including random sequence generation, allocation concealment, blinding of participants and personnel, blinding of outcome assessment, incomplete outcome data, selective reporting and other potential sources of bias.[21]

The reviewers will evaluate the seven risk of bias items one by one and grade the risk for each item as high, low or unclear. Disagreements will be resolved through discussion. A third reviewer (JZ) will be consulted to achieve consensus whenever necessary.

The authors of the original study will be contacted for further details if the information reported is insufficient to make a judgement.

Results of the assessment will be presented in a risk of bias summary figure and a risk of bias graph.[21]

### Measures of treatment effect

The risk ratio (RR) with its 95% CIs will be calculated for dichotomous outcomes.

The mean difference (MD) or standardised mean difference (SMD) with 95% CIs will be calculated for continuous outcomes.

The MD will be used if the same scale is used across different studies to measure an outcome. The SMD will be employed if different scales are used to measure the same outcome.

If a mixture of endpoint data and change from baseline data are reported for continuous outcomes in the included studies, subgroup analysis will be performed separately and the effect size estimates of different subgroups will not be pulled.[22]

A narrative description of the results will be provided if less than two trials are included for one outcome measure.[22]

### Unit of analysis issues

Only the data from the first phase of a crossover study will be analysed in this study.[23]

For studies dealing with multiple treatment groups, the review authors will make multiple pair-wise comparisons between pairs of groups of interest to the review question[21] and take care to avoid repeated inclusion of the same study group.

### Dealing with missing data

Data missing may occur for any outcome of interest. If necessary, we will contact authors of the original report for additional data. We will not fill missing data for any outcome in the primary analysis, but the impact of data missing will be assessed in the sensitivity analysis.

If the SD for an outcome is not reported, we will calculate the SD according to data reported or additional data collected from the authors.

Any potential impact of missing data will be discussed in the 'Discussion' section of this review.

### Assessment of heterogeneity

Statistical heterogeneity will be evaluated by visual inspection of the forest plot and the $X^2$ test. In addition, the $I^2$ statistical value will be calculated to quantify heterogeneity. Heterogeneity will be categorised according to the following rules.
► $I^2$ of 0%–40% as might not be important.

- $I^2$ of 30%–60% as may represent moderate heterogeneity.
- $I^2$ of 50%–90% as may represent substantial heterogeneity.
- $I^2$ of 75%–100% as considerable heterogeneity.

If the p value from a $X^2$ test is less than 0.10 or $I^2 > 50\%$, suggesting detectable between-study variation,[21 24] subgroup analysis considering prespecified factors will be performed in search of possible explanation for statistical heterogeneity.

### Assessment of reporting bias

Publication bias will be examined by assessing a funnel plot for signs of asymmetry if more than 10 studies are included in a meta-analysis.[23] If asymmetry is identified, possible explanations will be attempted.

### Data synthesis

An overall estimate of the intervention effect will be calculated by combining multiple study results in a meta-analysis on the condition that the same outcome of the same intervention was measured with comparable methods in a homogeneous population across studies.

The overall RR for dichotomous data will be estimated using the Mantel-Haenszel method.[23] For continuous data, the MD or SMD will be calculated in different situations.

When statistical heterogeneity is low, the fixed-effects model will be adopted. Otherwise, the random-effects model will be used to provide a more conservative estimate of the difference.

A descriptive summary of results from individual studies will be provided when it is impossible to conduct a meta-analysis.

### Subgroup analysis and investigation of heterogeneity

Subgroup analysis will be conducted for different pair of comparisons. The results of subgroup analysis will be presented in forest plots.

If one or two trials contributed to more than 80% of participants in a meta-analysis, the fixed-effects method will be preferred to provide a more conservative estimation.[25] Also, these studies will be analysed in one subgroup.[25]

A meta-regression analysis will be performed to investigate possible sources if heterogeneity is statistically significant ($p < 0.10$).

### Sensitivity analysis

We will perform sensitivity analysis to test the robustness of our findings taking into consideration the possible impact of the following factors:

- The model chosen for data pulling: to switch between fixed-effects model and random-effects model.
- Methodological quality of included studies: studies with high or unclear risk of bias will be excluded.
- Study design: studies with a crossover design will be excluded.

- Missing data: studies with over half of patients lost to follow-up will be excluded.

### Summary of findings table

Two review authors (MP and JZ) will independently assess the quality of evidence for each outcome using the Grading of Recommendations Assessment, Development and Evaluation (GRADE) approach. Five factors (study limitations, imprecision, inconsistency, indirectness and publication bias) could decrease the quality of evidence. The quality of evidence will fall into one of four categories from very low to high.[26]

A summary of findings table will be created including all the outcome measures using the GRADEpro guideline development tool.[27]

### Patient and public involvement

Patients were not involved in setting the research question or outcome measures, or in developing plans for the design or implementation of the study. There are no plans to disseminate the results of the research to study participants or any relevant patient community.

## AMENDMENTS

We will provide the date, description and rationale of any modification in the event of protocol amendments.

## DISSEMINATION

This systematic review will provide a comprehensive assessment of the efficacy and safety of probiotics, prebiotics or synbiotics for non-dialysis or non-renal transplant patients with CKD. The findings will be important for generating reliable recommendations for the clinical management of CKD.

We plan to publish systematic review findings in a peer-reviewed scientific journal and make the data sets openly accessible.

**Acknowledgements** The authors thank Wei Mu for revising the English language of the manuscript.

**Contributors** JZ, MP and HY conceived the study. HY provided general guidance for drafting the protocol. JZ and MP drafted the protocol. JS designed the search strategy. All author were involved in drafting, reviewing and revising the manuscript for intellectual content and have read and approved the final version of the manuscript.

**Funding** This study is supported by the National Natural Science Foundation of China (grant number 81703860), the National Natural Science Foundation of China (grant number 81673909) and Tianjin Science and Technology Program: Tianjin TCM Clinical Medicine Research Center (No. 15ZXLCSY00020).

**Competing interests** None declared.

**Patient consent** Not required.

**Provenance and peer review** Not commissioned; externally peer reviewed.

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
