## [Reviewer comments · BMJ Open]

ARTICLE DETAILS

TITLE (PROVISIONAL)	Probiotics, prebiotics and synbiotics for chronic kidney disease: protocol for a systematic review and meta-analysis
AUTHORS	Pei, Ming; Wei, Lijuan; Hu, Shouci; Yang, Bo; Si, Jinhua; Yang, Hongtao; Zhai, Jingbo

VERSION 1 – REVIEW

REVIEWER	Christiane Ishikawa Ramos Nutrition Programme, Universidade Federal de Sao Paulo, Brazil.
REVIEW RETURNED	20-Feb-2018

GENERAL COMMENTS	Dear authors The present systematic review and meta-analysis will access an important topic in chronic kidney disease: the manipulation of gut environment to improve microbial-derived uremic toxins. There is another registered protocol assessing pre-, pro- and synbiotics in CKD ongoing. However, differences in the primary outcome and some procedures would justify the occurrence of the present study. The current protocol has an adequate structure and procedures. I have minor comments/suggestions: Page 3, line 14 – to correct the word syNbiotics Page 5- I would suggest a change in the description of the inclusion criteria, regarding of the section “type of participants”. The term “any stage” might be confusing since patients under renal replacement therapy will be excluded. I understand that it is in line with the KDIGO classification but if you describe as “non-dialysis patients in any stage of CKD” (or as you would prefer, maintaining a consistence with other descriptions in the protocol) could be clearer. I would suggest specifying the fraction of uremic toxins are you looking for (free and total fractions?). Make sure that this description is consistent through the protocol. Kind regards
---

REVIEWER	Claire Trimmingham Central Northern Adelaide Renal and Transplantation Service, South Australia
REVIEW RETURNED	02-Mar-2018

GENERAL COMMENTS	Research question/objective Needs to clearly highlight that this review is looking at 'the effect that probiotics, prebiotics and synbiotics have on uremic toxin levels in CKD not dialysis'. The current title does not emphasise the focus of uremic toxins. Outcomes Given what is known in current literature you would expect that pro-,
--

	pre-, & synbiotics have an effect on gut health. By focusing on uremic toxins as the sole primary outcome you may be limiting the review. I would consider including GI symptoms, transit time, GI tolerance etc.
REVIEWER	Ana Paula Black Dreux Post Graduation Program in Medical Sciences, Fluminense Federal University (UFF), Niteroi-RJ, Brazil.
REVIEW RETURNED	04-Mar-2018
GENERAL COMMENTS	This research project is very well written, presenting all the necessary protocols for conducting a systematic review and meta-analysis. Because it was a research project, it was not possible to evaluate results, discussion and conclusion of the study, which will be evaluated later.

VERSION 1 – AUTHOR RESPONSE

Reviewers' Reports:

Reviewer: 1

Reviewer Name: Christiane Ishikawa Ramos

Institution and Country: Nutrition Programme, Universidade Federal de Sao Paulo, Brazil.

Competing Interests: None declared

Dear authors

The present systematic review and meta-analysis will access an important topic in chronic kidney disease: the manipulation of gut environment to improve microbial-derived uremic toxins. There is another registered protocol assessing pre-, pro- and synbiotics in CKD ongoing. However, differences in the primary outcome and some procedures would justify the occurrence of the present study. The current protocol has an adequate structure and procedures.

I have minor comments/suggestions:

Page 3, line 14 – to correct the word syNbiotics

Response:

Thank you for your suggestion. We have corrected the word on page 3.

Page 5- I would suggest a change in the description of the inclusion criteria, regarding of the section “type of participants”. The term “any stage” might be confusing since patients under renal replacement therapy will be excluded. I understand that it is in line with the KDIGO classification but if you describe as “non-dialysis patients in any stage of CKD” (or as you would prefer, maintaining a consistence with other descriptions in the protocol) could be clearer.

Response:

Thank you for your suggestion. We have made changes as required in the “Types of participants” section on page 4.

I would suggest specifying the fraction of uremic toxins are you looking for (free and total fractions?). Make sure that this description is consistent through the protocol.

Response:

Thank you for your suggestion. I have listed the fraction of uremic toxins in the “primary outcome” section on page 4. This description is consistent through the protocol.

Reviewer: 2

Reviewer Name: Claire Trimingham

Institution and Country: Central Northern Adelaide Renal and Transplantation Service, South Australia

Competing Interests: None declared

Research question/objective

Needs to clearly highlight that this review is looking at 'the effect that probiotics, prebiotics and synbiotics have on uremic toxin levels in CKD not dialysis'. The current title does not emphasise the focus of uremic toxins.

Response:

Thanks for this useful comment. I fully understood what you meant. We focus on multiple outcomes exhibiting the status of chronic kidney disease in this systematic review. Uremic toxins are considered the primary outcome, but other outcomes are also important. Therefore, uremic toxins were not emphasized in the title.

Outcomes

Given what is known in current literature you would expect that pro-, pre-, & synbiotics have an effect on gut health. By focusing on uremic toxins as the sole primary outcome you may be limiting the review. I would consider including GI symptoms, transit time, GI tolerance etc.

Response:

Thanks for your useful information. We have added these outcomes in the "Abstract" section on page 1 and the "secondary outcomes" section on page 4.

Reviewer: 3

Reviewer Name: Ana Paula Black Dreux

Institution and Country: Post Graduation Program in Medical Sciences, Fluminense Federal University (UFF), Niteroi-RJ, Brazil.

Competing Interests: 'None declared'

This research project is very well written, presenting all the necessary protocols for conducting a systematic review and meta-analysis.

Because it was a research project, it was not possible to evaluate results, discussion and conclusion of the study, which will be evaluated later.

Response:

Thanks for this useful comment. We will perform this systematic review according to the current protocol and discuss these topics in the final systematic review report.